# Improved regression in ratio type estimators based on robust M-estimation

**Khalid Ul Islam Rather**[ID][1]*, **Eda Gizem Koçyiğit**[ID][2], **Ronald Onyango**[3], **Cem Kadilar**[4]

**1** Division of Statistics and Computer Science, Chatha Jammu, India, **2** Department of Statistics, Dokuz Eylül University, Buca İzmir, Turkey, **3** Department of Applied Statistics, Financial Mathematics, and Actuarial Science, Jaramogi Oginga Odinga University of Science and Technology, Bondo, Kenya, **4** Department of Statistics, Hacettepe University, Beytepe, Ankara, Turkey

☬ These authors contributed equally to this work.
* khalidstat34@gmail.com

**Data Availability Statement:** All relevant data are within the paper.

**Funding:** The authors received no specific funding for this work.

## Abstract

In this article, a new robust ratio type estimator using the Uk's redescending M-estimator is proposed for the estimation of the finite population mean in the simple random sampling (SRS) when there are outliers in the dataset. The mean square error (MSE) equation of the proposed estimator is obtained using the first order of approximation and it has been compared with the traditional ratio-type estimators in the literature, robust regression estimators, and other existing redescending M-estimators. A real-life data and simulation study are used to justify the efficiency of the proposed estimators. It has been shown that the proposed estimator is more efficient than other estimators in the literature on both simulation and real data studies.

## Introduction

Outliers are observations that behave differently from the majority in datasets and often can significantly affect statistics. In sampling studies, however, the presence of outliers cannot be easily detected, since the entire population cannot always be reached. Especially in methods that need to work with small sample sizes, the efficiency of the estimation decreases if an outlier observation is taken into the sample.

To reduce the consequences of an outlier(s) in the real data, robust regression methods are generally used. M-estimators are used as a robust replacement for the general classical estimators utilized in the field of statistics. To overcome the problem of outliers efficiently as compared to other robust estimation methods, the Uk's redescending M-estimator is proposed [1]. The outliers in the data mainly affect the traditional estimation methods and reduce their efficiencies. In fact, the performance of the ordinary least square (OLS) estimators reduces in the presence of outliers. Therefore, numerous redescending M-estimation methods have been developed to control the consequences of outliers and to improve the efficiency of the procedures, including [2–30]. This study aims to reduce the effect of outliers by developing a new ratio-type redescending M-estimator based on the Uk's redescending M-estimator (URME) that may improve the efficiency of URME and provide a perfect estimation.

**Competing interests:** The authors have declared that no competing interests exist.

This article is organized as follows: Section 2 introduces the traditional ratio estimators based on previous estimators and some existing M-estimators in the literature. In Section 3, we give brief information about Uk's redescending M-Estimator and then present the proposed estimator. In addition, efficiency comparisons of the proposed estimator are given in the last part of this section. Section 4 calculates the relative efficiencies of the estimators and compares these estimators with each other in theory and in the application by simulation and real data, respectively. Lastly, Section 5 concludes and offers for the future studies.

## Existing estimators in the literature

### Kadilar and Cingi (2004) estimators

In the simple random sampling, Kadilar and Cingi [31] introduced ratio estimators by adapting the traditional estimators and other ratio-type estimators in literature [32]. On the basis of MSE equations and numerical illustrations, it was proved that the efficiencies of the proposed estimators are better than OLS estimators. These estimators are

$$\overline{y}_{KC_j} = \frac{\overline{y} + b(\overline{X} - \overline{x})}{(\beta_j \overline{x} + \gamma_i)} (\beta_j \overline{X} + \gamma_i), j = 1, 2, 3, 4, 5 \tag{1}$$

where $b$ is the slope coefficient derived from the OLS estimation, $\overline{y}$ is the observed sample mean of the study variable and $\overline{x}$ is the sample mean of the auxiliary variable. Also, $\beta_1 = 1$ and $\gamma_1 = 0, \beta_2 = 1$ and $\gamma_2 = C_x, \beta_3 = 1$ and $\gamma_3 = \beta_2(x), \beta_4 = \beta_2(x)$ and $\gamma_4 = C_x, \beta_5 = C_x$ and $\gamma_5 = \beta_2(x)$. Here, $\beta_2(x)$ and $C_x$ are both the population coefficient of kurtosis and coefficient of variation of the auxiliary variable, respectively. It should be noted that, when we do not have the population parameters, we can estimate these parameters from the sample. The MSE equation of $\overline{y}_{KC_j}$ is as follows:

$$MSE(\overline{y}_{KC_j}) \cong \theta(R_j^2 S_x^2 + 2BR_j S_x^2 + B^2 S_x^2 - 2R_j S_{xy} - 2BS_{xy} + S_y^2) \text{ for } j = 1, 2, 3, 4, 5 \tag{2}$$

where $\theta = \frac{1-f}{n}, f = \frac{n}{N}$ and $B$ is obtained by an expected value of $b$ such that $E(b) = B$.

$$R_j = \frac{\beta_j \overline{Y}}{\beta_j \overline{X} + \gamma_j} \text{ for } j = 1, 2, 3, 4, 5 \tag{3}$$

It is worth to note that the ratio estimator, given in Eq (1), has higher potentiality and proficiency in the existence of outliers than that of other traditional estimators in the literature [33,34]. However, the occurrence of outliers vanishes the productivity and proficiency of these estimators. Therefore, Kadilar *et al.* [35] proposed new ratio estimators for the efficient estimation of the population mean.

### Kadilar et al. (2007) [35] Huber M-estimators

For the regression analysis, numerous methods have been introduced in the literature to deal with the problem of outliers in the data. Such estimators were initially developed by Huber [9], but later on, Kadilar *et al.* [35] gave emphasis on these estimators by using the robust regression as a substitute for OLS. The estimators were named as Huber M-estimators (HM) and they were given as

$$\overline{y}_{HM_j} = \frac{\overline{y} + b_{HM}(\overline{X} - \overline{x})}{(\beta_j \overline{x} + \gamma_i)} (\beta_j \overline{X} + \gamma_i), j = 1, 2, 3, 4, 5 \tag{4}$$

where $b_{HM}$ is the slope coefficient of the robust regression M-estimators given by Huber [9].

The design of Huber's function $\rho(r_j)$ is given by

$$\rho(r_j) = \begin{cases} \dfrac{r^2}{2} & for\,|r| \leq c \\[2mm] c|r| - \dfrac{c^2}{2} & for\,|r| > c \end{cases} \tag{5}$$

where $r$ is the random error following the OLS method while $c$ is the tuning constant.

The advised value of $c$ from the Huber [9] is one and half times of the estimated standard deviation of error. The MSE equation of the M-estimators is given as follows:

$$MSE(\overline{y}_{HM_j}) \cong \theta(R_j^2 S_x^2 + 2B_{HM}R_j S_x^2 + B_{HM}^2 S_x^2 - 2R_j S_{xy} - 2B_{HM}S_{xy} + S_y^2)$$
$$for\ j = 1, 2, 3, 4, 5 \tag{6}$$

where $B_{HM}$ is the expected slope coefficient of $b$. The MSE for the estimators, given in Eq (6), can also be obtained by replacing $B$ in Eq ((2) with $B_{HM}$. The MSE computed for M-estimators are relatively more efficient as compared to the OLS estimators.

## Raza et al. (2019) [36] estimators

Raza *et al.* [36] proposed ratio estimators based on the newly developed robust redescending M-estimator. The redescending M-estimators (RM) are given by

$$\overline{y}_{RM_j} = \frac{\overline{y} + b_{RM}(\overline{X} - \overline{x})}{(\beta_j \overline{x} + \gamma_i)}(\beta_j \overline{X} + \gamma_i),\ j = 1, 2, 3, 4, 5 \tag{7}$$

where $b_{RM}$ is the slope coefficient of the redescending M-estimators given by Raza *et al.* [36]. The design of the Raza's objective function $\rho(r_j)$ is described as

$$\rho_1(r_j) = \frac{v^2}{2c}\left\{1 - \left[1 + \left(\frac{r}{v}\right)^2\right]^{-c}\right\} for\ |r| \geq 0 \tag{8}$$

where $c$ and $v$ are tuning constants. For the current study, optimum values of the tuning constant are $c = 2.5$ and $v = 8$. The $b_{RM}$ redescending M-estimator is used in the MSE equation of the ratio estimators in Eq (7) as follows:

$$MSE(\overline{y}_{RM_j}) \cong \theta(R_j^2 S_x^2 + 2B_{RM}R_j S_x^2 + B_{RM}^2 S_x^2 - 2R_j S_{xy} - 2B_{RM}S_{xy} + S_y^2)$$
$$for\ j = 1, 2, 3, 4, 5 \tag{9}$$

## Noor-ul-Amin et al. (2020) [37] estimators

Noor-ul-Amin *et al.* [37] proposed another ratio estimator using the robust M-estimators and named it as redescending M-estimators under the different objective function given by

$$\overline{y}_{NM_j} = \frac{\overline{y} + b_{NM}(\overline{X} - \overline{x})}{(\beta_j \overline{x} + \gamma_i)}(\beta_j \overline{X} + \gamma_i),\ j = 1, 2, 3, 4, 5 \tag{10}$$

where $b_{NM}$ is the slope coefficient of the redescending M-estimators given by Noor-ul-Amin *et al.* [37]. The design of the Noor objective function $\rho_2(r_j)$ is described as

$$\rho_2(r_j) = \frac{c^2}{4}\left[\frac{\tan^{-1}\left(\frac{2r}{c}\right)^2}{4} + \frac{r^2 c^2}{c^4 + 16r^4}\right] for\ |r| \geq 0 \tag{11}$$

The MSE equation of the ratio estimators, given in Eq (10), is calculated with the same method as that given before.

$$MSE(\bar{y}_{NM_j}) \cong \theta(R_j^2 S_x^2 + 2B_{NM}R_j S_x^2 + B_{NM}^2 S_x^2 - 2R_j S_{xy} - 2B_{NM}S_{xy} + S_y^2)$$

$$\text{for } j = 1, 2, 3, 4, 5 \tag{12}$$

## Proposed ratio estimators based on Uk's redescending M-estimator

### Uk's redescending M-estimator

The proposed estimator is also known as Uk's redescending M-estimator. The M-estimator of $\beta$ is defined by the following objective function:

$$Minimize\hat{\beta}\sum_{i=1}^{n}\rho(r_i) \tag{13}$$

where $r_i = y_i - \beta x_i$ represents the residuals. An objective function must fullfill the following standard properties:

- $\rho(0) = 0$

- $\rho(r_i) \geq 0$

- $\rho(r_i) = \rho(-r_i)$

- $\rho(r_i) \geq \rho(r_j)$ for $|r_i| \geq |r_j|$

- $\rho$ is differentiable

M-estimator is called a redescending M-estimator if it fullfils the standard related properties and the derivative of its objective function is $\psi$-function. Differentiating Eq (13) with respect to $\hat{\beta}_j$ we obtain $\psi(r_i)$ function as follows:

$$\sum_{i=1}^{n}\psi(r_i)X_i = 0 \tag{14}$$

Dividing $\psi(r_i)$ by $r_i$ we obtain the weight function as

$$\sum_{i=1}^{n}w(r_i)X_i = \frac{\sum_{i=1}^{n}\psi(r_i)X_i}{r} \tag{15}$$

On the base of procedure, defined in Eqs (5), (8) and (11), a redescending M-estimator is proposed with the aid of [1]. The objective function of the proposed estimator is

$$\rho(r) = \begin{cases} \frac{3}{2}\sin\left(\frac{4}{9}\right)\left[\frac{r^{10}}{10c^8} - \frac{r^6}{3c^4} + \frac{r^2}{2}\right] & for \ |r| \leq c \\ \frac{3}{2}\sin\left(\frac{16c^2}{135}\right) & for \ |r| \leq c \end{cases} \tag{16}$$

Differentiating Eq ([16]) w.r.t $\hat{\beta}_j$ we get the $\psi$-function as

$$\psi(r) = \begin{cases} r\left(\dfrac{3}{2}\right)\left[1 - \left(\dfrac{r}{c}\right)^4\right]^2 \sin\left\{\dfrac{2}{3}\left[1 - \left(\dfrac{r}{c}\right)^4\right]^2\right\} & \text{for } |r| \leq c \\ 0 & \text{for } |r| \leq c \end{cases} \tag{17}$$

Dividing $\psi(r_i)$ by residual, we obtain the weight function as

$$w(r) = \begin{cases} \left(\dfrac{3}{2}\right)\left[1 - \left(\dfrac{r}{c}\right)^4\right]^2 \sin\left\{\dfrac{2}{3}\left[1 - \left(\dfrac{r}{c}\right)^4\right]^2\right\} & \text{for } |r| \leq c \\ 0 & \text{for } |r| \leq c \end{cases} \tag{18}$$

The graphs of the objective $\rho$-function, $\psi$-function, and weight function are shown in Fig 1A–1C, respectively.

## Proposed estimator

Motivated from the estimators [31,35,36,37] in literature and by using the Uk's Redescending M-Estimator [1], the proposed estimator is defined as follows:

$$\overline{y}_{UK_i} = \frac{\overline{y} + b_{UK}(\overline{X} - \overline{x})}{(\beta_j \overline{x} + \gamma_i)}(\beta_j \overline{X} + \gamma_i), i = 1, 2, 3, 4, 5 \tag{19}$$

The MSE equation of the estimator in the Eq ([19]) is obtained by

$$MSE(\overline{y}_{UK_i}) \cong \theta(R_j^2 S_x^2 + 2B_{UK}R_j S_x^2 + B_{UK}^2 S_x^2 - 2R_j S_{xy} - 2B_{UK}S_{xy} + S_y^2)$$
$$\text{for } i = 1, 2, 3, 4, 5 \tag{20}$$

where $B_{UK}$ is calculated from the objective function in the Eq ([16]) and $R_1 = \overline{Y}/\overline{X}$
$R_2 = \overline{Y}/(\overline{X} + C_x)$, $R_3 = \overline{Y}/(\overline{X} + \beta_2(x))$, $R_4 = \beta_2(x)\overline{Y}/(\beta_2(x)\overline{X} + C_x)$, and
$R_5 = C_x\overline{Y}/(C_x\overline{X} + \beta_2(x))$.

To evaluate the efficiency of the proposed ratio estimator, MSE equations of the estimators will be compared in Section 3.3.

## Efficiency comparisons

For the theoretical comparisons of the proposed estimator with other estimators, it is first necessary to compare it with the traditional estimator proposed by Kadilar and Cingi [31].

$$MSE(\overline{y}_{KC_j}) > MSE(\overline{y}_{UK_i})$$

$$2BR_j S_x^2 + B^2 S_x^2 - 2BS_{xy} - 2B_{UK}R_j S_x^2 - B_{UK}^2 S_x^2 + 2B_{UK}S_{xy} > 0$$

$$2R_j(B - B_{UK}) + (B^2 - B_{UK}^2) - 2b(B - B_{UK}) > 0$$

$$(B - B_{UK})(2R_j - 2b) + (B - B_{UK})(B + B_{UK}) > 0$$

where $b$ is LS slope obtained by the OLS method.

$$(B - B_{UK})(2R_j + B + B_{UK} - 2b) > 0 \tag{21}$$

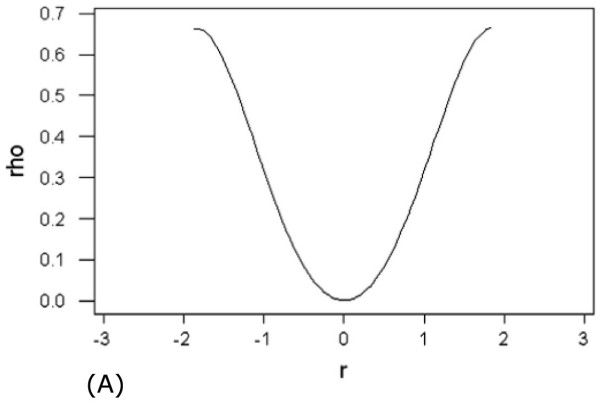

(A)

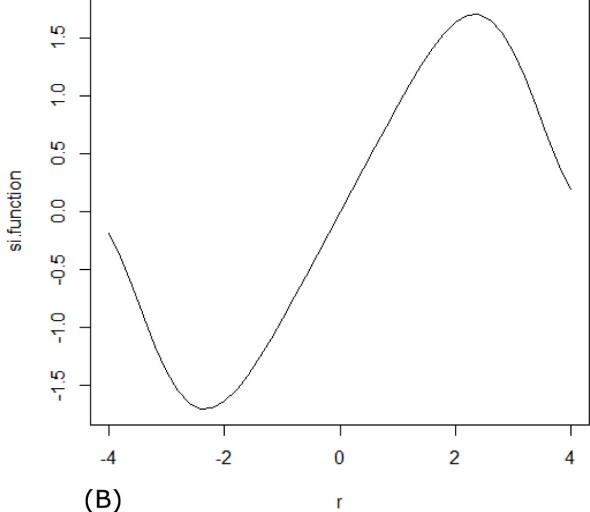

(B)

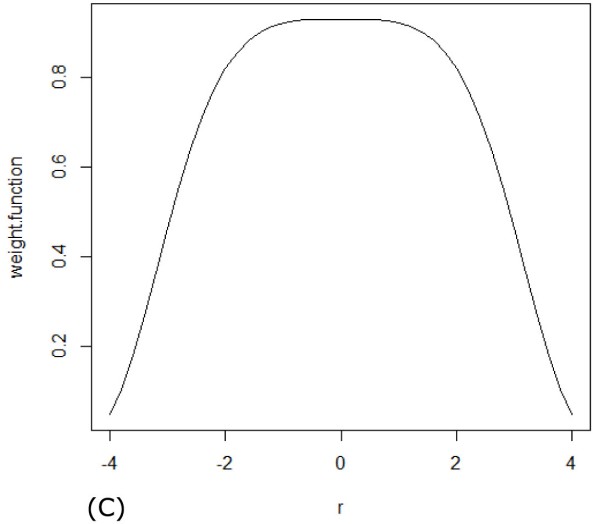

(C)

**Fig 1.** Graphs of the functions of Uk's M-estimator (A) Objective function, (B) Ψ-function and (C) weight function.

From Eq (21), it is possible to compare the estimators to a general formula with $B^*$ which can be $B, B_{HM}, B_{RM}$, and $B_{NM}$ as follows:

1. $B^* > B_{UK}$

2. $B^* + B_{UK} > 2(b - R_j)$

If the given Conditions (I) and (II) are satisfied, the proposed estimator is the most efficient estimator.

## Numerical comparisons

### Real data studies

To prove the efficiency of the proposed estimators, real-life data examples are considered. For this strategy, we use two different datasets. The first real dataset is the apple production data taken from the Black Sea Region in Turkey [35]. Apple production in tons is taken as a study variable and the number of trees (1 unit = 1000 trees) in 204 villages is taken as an auxiliary variable. Table 1 shows the population parameters for the first real dataset. Fig 2 shows the scatter plot of the data where outliers can be seen clearly.

For the comparison, the reference estimator is $\bar{y}_{KC_i}$ traditional ratio estimator. Percent relative efficiency is computed by using

$$PRE(\bar{y}_{pi}) = \frac{MSE(\bar{y}_{KC_j})}{MSE(\bar{y}_{*_j})} * 100; \ j = 1, 2, 3, 4, 5 \tag{22}$$

where $* = HM, RM, NM$ and $UK$. 10000 sample size of n = 30 were drawn from the population which is size N = 204 and the PREs were calculated using Eq (22) and the values obtained are given in Table 2. The best predictors are marked with "*" in the table.

The second real dataset concerning the U.S. State Public-School Expenditures is used. This data consists of fifty-one observations indicating the per-capita income in dollars and per-capita education expenditure in dollars for the U. S. states in 1970 [38]. The per-capita income is taken as the study variable and per-capita education expenditure is taken as an auxiliary variable. The original data was free from outliers. For this reason, a 7% outlier was added as in Raza [36]. The scatter plots of the original and outlier-added data are given within the Fig 3. The parameters of each population are given in Table 3. All of the calculations have been made as in the first real dataset and the obtained PRE values are given in Table 4. The best estimators are marked with "*". As shown in Table 4, we see that the proposed estimators are quite efficient estimators according to other estimators, especially for the outlier-added data.'

A comparison of the proposed estimators with each other for all real datasets used is summarized in Fig 4. Accordingly, it can be inferred that among the proposed estimators, $\bar{y}_{UK5}$ is the most effective one in general.

**Table 1. Parameters of apple production dataset.**

| | | |
|---|---|---|
| N = 204 | $\beta_2(x) = 29.77$ | $\rho = 0.713$ |
| n = 30 | $C_x = 1.717$ | $B = 4.165872$ |
| $\bar{Y} = 966.96$ | $R_{KC1} = 3.656933$ | $B_{KC} = 3.556434$ |
| $\bar{X} = 264.42$ | $R_{KC2} = 3.633339$ | $B_{NM} = 2.50765$ |
| $S_y = 2389.77$ | $R_{KC3} = 3.286817$ | $B_{UK} = 2.497468$ |
| $S_x = 454.03$ | $R_{KC4} = 3.656136$ | |
| $S_{xy} = 773727.8$ | $R_{KC5} = 3.431872$ | |

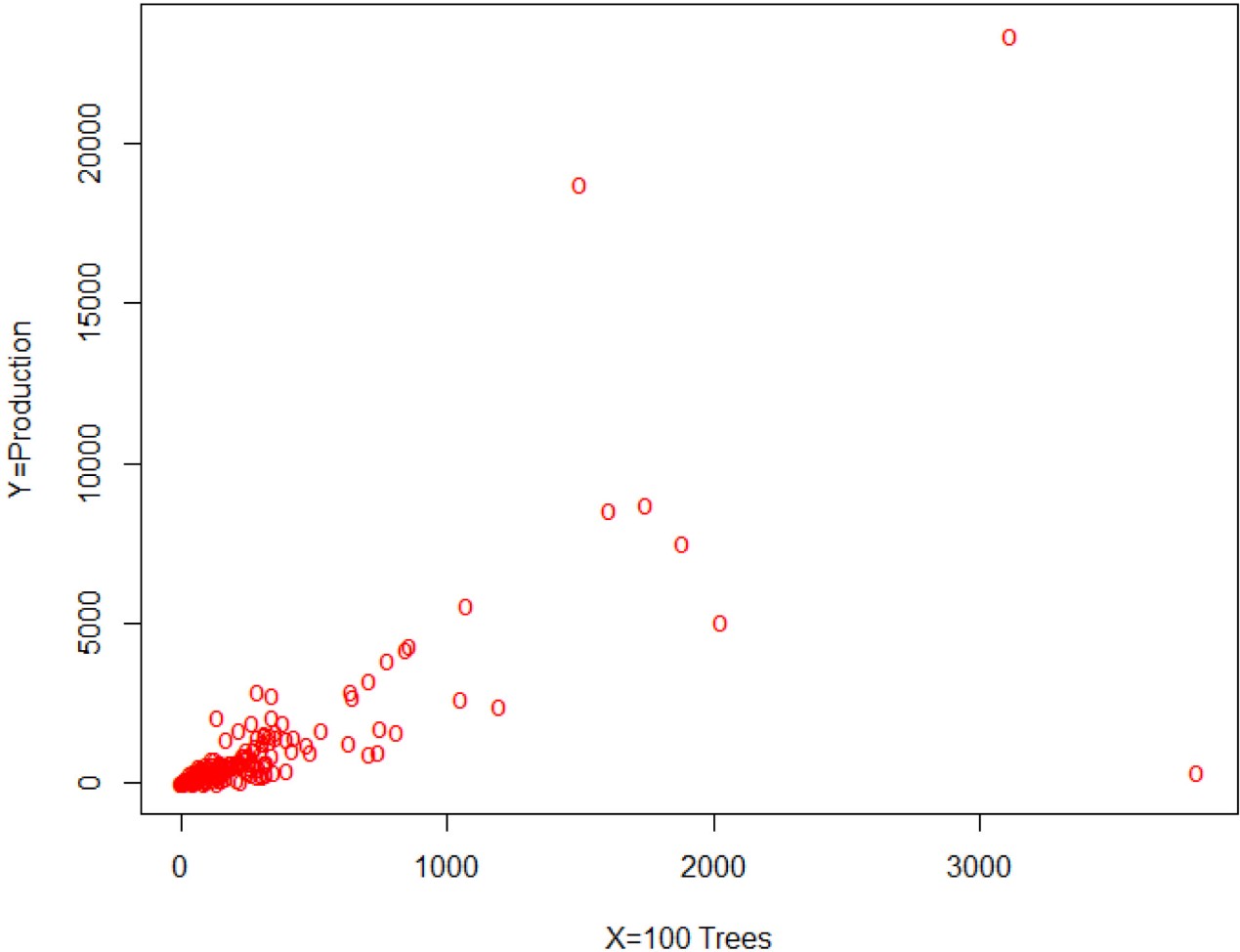

**Fig 2. Scatter plot of apple dataset.**

In all of the various real datasets used, the proposed estimator is found to be the most efficient estimator. Theoretically, for Condition (I), it can be seen from Tables 1 and 3 that the

**Table 2. PREs for the apple production dataset (%).**

| Reference | $\bar{y}_{HMj}$ | $\bar{y}_{NMj}$ | $\bar{y}_{UKj}$ |
|---|---|---|---|
| $\bar{y}_{KC1}$ | 125.0948 | 144.6571 | 145.0107* |
| $\bar{y}_{KC2}$ | 125.1424 | 144.3392 | 144.6912* |
| $\bar{y}_{KC3}$ | 125.5125 | 142.014 | 142.3519* |
| $\bar{y}_{KC4}$ | 125.0969 | 144.629 | 144.9825* |
| $\bar{y}_{KC5}$ | 125.2602 | 142.9627 | 143.3072* |

**Original Data**

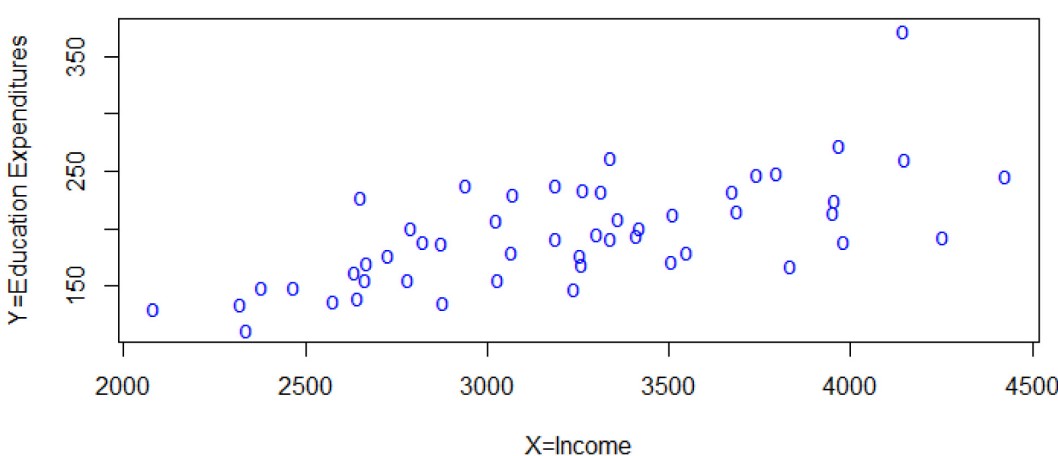

**Data with Outlier**

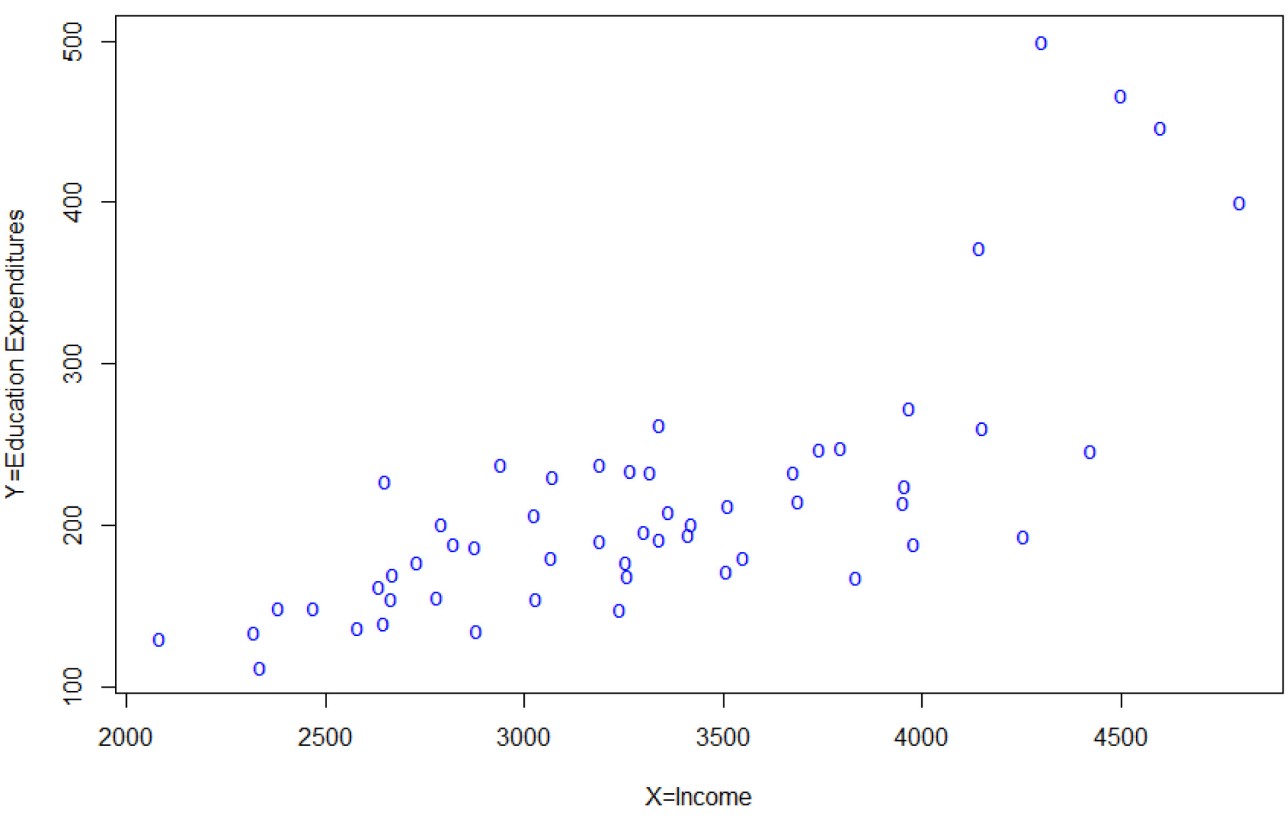

**Fig 3. Scatter plot of public school expenditures dataset.**

**Table 3. Parameters of public school expenditures real dataset.**

| Original | N = 51<br>n = 30<br>$\overline{Y}$ = 196.3137<br>$\overline{X}$ = 3225.294<br>$S_y$ = 46.45449<br>$S_x$ = 560.026<br>$S_{xy}$ = 17367.51 | $\beta_2(x)$ = 2.288739<br>$C_x$ = 0.1736356<br>$R_{KC1}$ = 0.06086692<br>$R_{KC2}$ = 0.06086365<br>$R_{KC3}$ = 0.06082376<br>$R_{KC4}$ = 0.06086549<br>$R_{KC5}$ = 0.06061918 | $\rho$ = 0.6675773<br>B = 0.05537594<br>$B_{KC}$ = 0.05353533<br>$B_{NM}$ = 0.05739941<br>$B_{UK}$ = 0.05033515 |
|---|---|---|---|
| Outlier-added | N = 55<br>n = 30<br>$\overline{Y}$ = 215.0182<br>$\overline{X}$ = 3321.618<br>$S_y$ = 81.4779<br>$S_x$ = 642.8723<br>$S_{xy}$ = 38997.47 | $\beta_2(x)$ = 2.40009<br>$C_x$ = 0.1935419<br>$R_{KC1}$ = 0.06473296<br>$R_{KC2}$ = 0.06472919<br>$R_{KC3}$ = 0.06468622<br>$R_{KC4}$ = 0.06473139<br>$R_{KC5}$ = 0.06449219 | $\rho$ = 0.7445123<br>B = 0.09346485<br>$B_{KC}$ = 0.07535126<br>$B_{NM}$ = 0.05532234<br>$B_{UK}$ = 0.05060727 |

BUK value is lower than the other B values. The information given in Tables 1 and 3 also shows that Condition (II) of Eq (21) is satisfied in Table 5.

## Simulation study

The simulation study is also conducted to check the superiority of the proposed estimator. For this purpose, data is generated from the normal distribution for representing symmetric distributions and exponential distribution for skewed distributions by using the R software. Results are calculated from the 10000 SRS (without replacement) samples. Efficiency is compared for 20, 30, 40, and 50 sample sizes of *n*. Also, we consider the outlier rates as 0.05 and 0.1. The following regression model is used to generate data for the simulation study:

$$y_i = \alpha + bx_i + e_i$$

where $e_i$ refers to residuals and $\alpha$ = 2, b = 1.

To verify the efficiency of the proposed estimator, 95% of the study variable is generated using $N(20,10)$, and 5% of the variable is generated using $N(50,10)$ for outlier data. Similarly, for the skewed distribution, 95% of the study variable is generated using Exp(3), and 5% of the variable is generated using Exp(15) for outlier data. Residuals are generated using the same ratio of N(0,1) with N(30,1), and Exp(1) with Exp(5) respectively. The tuning constants were taken as 1.5 for Huber, and 3 for NM and UK as suggested. Note that this simulation study is repeated for 10% outlier data as well. The calculated *B* coefficients are given in Table 6 for both distribution. PRE values were calculated using Eq (22) and the results are given in Table 7. The

**Table 4. PREs for the public school expenditures real dataset (%).**

| Reference | Original | | | Outlier of 7% | | |
|---|---|---|---|---|---|---|
| | $\overline{y}_{HMj}$ | $\overline{y}_{NMj}$ | $\overline{y}_{UKj}$ | $\overline{y}_{HMj}$ | $\overline{y}_{NMj}$ | $\overline{y}_{UKj}$ |
| $\overline{y}_{KC1}$ | 104.4891 | 97.94668 | 104.8218* | 125.3495 | 146.8591 | 151.5663* |
| $\overline{y}_{KC2}$ | 104.4891 | 97.94663 | 104.8218* | 125.349 | 146.8564 | 151.563* |
| $\overline{y}_{KC3}$ | 104.4891 | 97.94605 | 104.8212* | 125.3436 | 146.8273 | 151.5265* |
| $\overline{y}_{KC4}$ | 104.4891 | 97.94666 | 104.8218* | 125.3492 | 146.8579 | 151.5649* |
| $\overline{y}_{KC5}$ | 104.4892 | 97.94336 | 104.8185* | 125.3182 | 146.697 | 151.3624* |

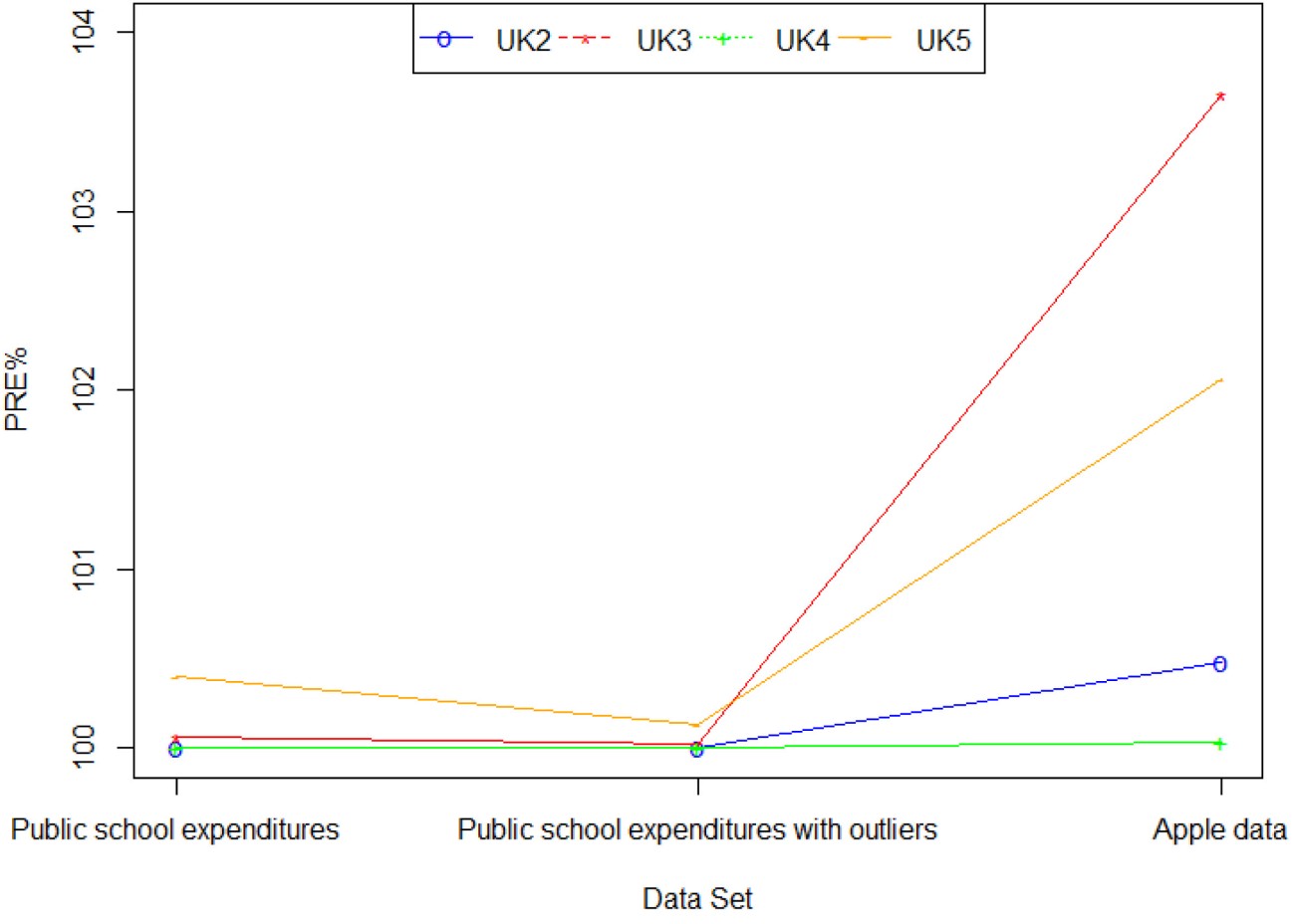

**Fig 4. Comparison plot of the proposed estimators.**

best predictors are marked as before. Note that the PRE values of the proposed estimators are also presented in Fig 5 for both distribution.

## Conclusion

In the simple random sampling, under the determined conditions, the ratio estimators are employed to estimate the population mean efficiently. On the other hand, M-estimators are developed in the case that the data contain outliers. It has been seen from the studies in the literature that more effective results are obtained as a result of combining the ratio estimators and the M-estimators in the presence of outliers. Our results require additional precision; however, the outliers violate the OLS assumptions and do not produce good results. We present a Uk's redescending M-estimator-based ratio estimator to solve this problem. To support

**Table 5. Control of condition II for efficiency of the proposed estimator.**

| Real Data Sets | $R_i$ | $2(b\text{-}R_i)$ | $B^*+B_{UK}$ | |
|---|---|---|---|---|
| Apple data set | 1 | 0.1929 | $B+B_{UK}$ | 6.6633 |
| | 2 | 0.2401 | $B_{KC}+B_{UK}$ | 5.0051 |
| | 3 | 0.9330 | $B_{NM}+B_{UK}$ | 5.0051 |
| | 4 | 0.1945 | | |
| | 5 | 0.6430 | | |
| Public school expenditures original data set | 1 | -0.01099 | $B+B_{UK}$ | 0.1057 |
| | 2 | -0.01098 | $B_{KC}+B_{UK}$ | 0.1039 |
| | 3 | -0.0109 | $B_{NM}+B_{UK}$ | 0.1077 |
| | 4 | -0.01098 | | |
| | 5 | -0.01049 | | |
| Public school expenditures data set with outlier | 1 | 0.05925 | $B+B_{UK}$ | 0.1441 |
| | 2 | 0.05926 | $B_{KC}+B_{UK}$ | 0.12596 |
| | 3 | 0.05935 | $B_{NM}+B_{UK}$ | 0.1059 |
| | 4 | 0.05926 | | |
| | 5 | 0.05974 | | |

the proposed estimators, real-life data examples and a simulation study are conducted and they prove the efficiency of the proposed estimator.

In real data studies, it is noteworthy that the proposed estimators are more effective than the others. It was observed that the efficiency of robust estimators increased as the number of outliers increased in the data. The most striking point observed in real data studies is on the original public school expenditures real dataset. The efficiency of the $\overline{y}_{NMj}$ estimators on this real dataset without outliers is even lower than the reference that is a non-robust ratio estimator. In contrast, the proposed estimator is still the most efficient estimator.

The simulation results are also obtained in a way that supports the real data study. As the number of outliers increases, the efficiency of robust estimators increases and the most effective one is the proposed estimators again. It was observed that the efficiency in the skewed

**Table 6. $B$ coefficients for M-Estimators under different distributions, rate of outliers and sample sizes.**

| | Normal Distribution | | | | | | | |
|---|---|---|---|---|---|---|---|---|
| $n$ | Outlier Rate: 5% | | | | Outlier Rate: 10% | | | |
| | $B$ | $B_{HM}$ | $B_{NM}$ | $B_{UK}$ | $B$ | $B_{HM}$ | $B_{NM}$ | $B_{UK}$ |
| 20 | 1.319016 | 1.026611 | 0.990509 | 0.982837 | 1.473978 | 1.136479 | 1.014438 | 1.002837 |
| 30 | 1.239935 | 1.006161 | 0.996079 | 0.992051 | 1.463691 | 1.109682 | 1.011521 | 1.002521 |
| 40 | 1.308943 | 0.992054 | 0.983044 | 0.979714 | 1.459319 | 1.096287 | 1.009964 | 1.002419 |
| 50 | 1.266481 | 1.012449 | 1.003065 | 1.002283 | 1.456715 | 1.089494 | 1.008753 | 1.002176 |
| | Exponential Distribution | | | | | | | |
| $n$ | Outlier Rate: 5% | | | | Outlier Rate: 10% | | | |
| | $B$ | $B_{HM}$ | $B_{NM}$ | $B_{UK}$ | $B$ | $B_{HM}$ | $B_{NM}$ | $B_{UK}$ |
| 20 | 1.593572 | 1.395416 | 1.19901 | 0.9389895 | 1.722707 | 1.511503 | 1.301762 | 1.047658 |
| 30 | 1.447282 | 1.266454 | 1.132652 | 0.9385277 | 1.568694 | 1.384954 | 1.266967 | 1.064343 |
| 40 | 1.397501 | 1.236879 | 1.146027 | 0.981304 | 1.490443 | 1.331446 | 1.226918 | 1.067527 |
| 50 | 1.34828 | 1.194397 | 1.108079 | 0.9730708 | 1.447567 | 1.29981 | 1.217351 | 1.07748 |

**Table 7. PREs of robust estimators by simulation (%).**

| | $n$ | Reference | Outlier Rate: 5% | | | Outlier Rate: 10% | | |
|---|---|---|---|---|---|---|---|---|
| | | | $\overline{y}_{HMj}$ | $\overline{y}_{NMj}$ | $\overline{y}_{UKj}$ | $\overline{y}_{HMj}$ | $\overline{y}_{NMj}$ | $\overline{y}_{UKj}$ |
| Normal Dist. | 20 | $\overline{y}_{KC1}$ | 164.6673 | 169.9909 | 173.6181* | 173.1882 | 218.059 | 222.5642* |
| | | $\overline{y}_{KC2}$ | 166.6329 | 172.1144 | 175.8659* | 175.3731 | 221.7894 | 226.4527* |
| | | $\overline{y}_{KC3}$ | 175.278 | 180.8811 | 185.0921* | 186.0025 | 236.7097 | 241.8854* |
| | | $\overline{y}_{KC4}$ | 165.271 | 170.6599 | 174.3265* | 173.7791 | 219.1975 | 223.7529* |
| | | $\overline{y}_{KC5}$ | 180.9139 | 186.540 | 190.969* | 192.1066 | 244.8612 | 250.1742* |
| | 30 | $\overline{y}_{KC1}$ | 147.2726 | 149.2286 | 151.9068* | 181.889 | 218.3278 | 221.8743* |
| | | $\overline{y}_{KC2}$ | 148.6053 | 150.6146 | 153.3773* | 184.3227 | 221.9861 | 225.6531* |
| | | $\overline{y}_{KC3}$ | 154.7781 | 156.8499 | 159.9852* | 196.397 | 238.1077 | 242.2264* |
| | | $\overline{y}_{KC4}$ | 147.6858 | 149.664 | 152.3682* | 182.5465 | 219.3851 | 222.9674* |
| | | $\overline{y}_{KC5}$ | 159.5052 | 161.6217 | 165.0045* | 203.9167 | 247.6544 | 251.9397* |
| | 40 | $\overline{y}_{KC1}$ | 163.6049 | 169.3674 | 171.8475* | 186.5542 | 218.4778 | 221.4503* |
| | | $\overline{y}_{KC2}$ | 165.4164 | 171.3515 | 173.9109* | 189.1126 | 222.084 | 225.1561* |
| | | $\overline{y}_{KC3}$ | 174.9103 | 181.4948 169.9191 188.6655 | 184.427* | 201.9794 | 238.766 | 242.2344* |
| | | $\overline{y}_{KC4}$ | 164.1029 | | 172.4215* | 187.2422 | 219.4918 | 222.493* |
| | | $\overline{y}_{KC5}$ | 181.7143 | | 191.8099* | 210.3214 | 249.0572 | 252.6816* |
| | 50 | $\overline{y}_{KC1}$ | 153.6796 | 157.5499 | 159.5689* | 188.9728 | 218.8553 | 221.4482* |
| | | $\overline{y}_{KC2}$ | 155.1604 | 159.1438 | 161.2252* | 191.5885 | 222.4366 | 225.1154* |
| | | $\overline{y}_{KC3}$ | 163.0814 | 167.5228 | 169.9198* | 204.9737 | 239.6492 | 242.6888* |
| | | $\overline{y}_{KC4}$ | 154.0893 | 157.9946 | 160.031* | 189.6669 | 219.8365 | 222.4534* |
| | | $\overline{y}_{KC5}$ | 169.1521 | 173.8924 | 176.4997* | 213.8642 | 250.5479 | 253.7377* |
| Exp. Distr. | 20 | $\overline{y}_{KC1}$ | 103.0984 | 106.3664 | 111.3753* | 103.2159 | 106.6792 | 111.4581* |
| | | $\overline{y}_{KC2}$ | 105.7525 | 108.2843 | 117.6538* | 106.0368 | 108.8238 | 118.0747* |
| | | $\overline{y}_{KC3}$ | 103.0054 | 99.91533 | 103.591* | 103.0935 | 100.0658 | 103.8029* |
| | | $\overline{y}_{KC4}$ | 104.3554 | 108.8531 | 116.6138* | 104.6074 | 109.4754 | 117.0385* |
| | | $\overline{y}_{KC5}$ | 102.8503 | 99.77669 | 103.3443* | 102.9868 | 100.0501 | 103.753* |
| | 30 | $\overline{y}_{KC1}$ | 103.0305 | 105.3887 | 109.1184* | 103.0083 | 104.9156 | 108.7313* |
| | | $\overline{y}_{KC2}$ | 105.5871 | 108.2079 | 115.1546* | 105.6728 | 107.3196 | 114.8008* |
| | | $\overline{y}_{KC3}$ | 102.5481 | 101.1733 | 103.4809* | 102.4865 | 100.5953 | 103.2919* |
| | | $\overline{y}_{KC4}$ | 104.1721 | 107.4634 | 112.9505* | 104.2532 | 106.8446 | 112.6375* |
| | | $\overline{y}_{KC5}$ | 102.4661 | 101.0941 | 103.3453* | 102.4833 | 100.6456 | 103.376* |
| | 40 | $\overline{y}_{KC1}$ | 102.7439 | 104.2812 | 107.4394* | 102.6497 | 104.3912 | 107.4007* |
| | | $\overline{y}_{KC2}$ | 105.1112 | 106.8234 | 112.8136* | 105.0196 | 107.0448 | 112.8866* |
| | | $\overline{y}_{KC3}$ | 102.0964 | 101.0672 | 103.0307* | 102.0106 | 100.9452 | 102.8198* |
| | | $\overline{y}_{KC4}$ | 103.7787 | 105.8661 | 110.4337* | 103.6701 | 106.0532 | 110.4785* |
| | | $\overline{y}_{KC5}$ | 102.0616 | 101.0429 | 102.9948* | 102.0377 | 101.0163 | 102.95* |
| | 50 | $\overline{y}_{KC1}$ | 102.6618 | 104.1679 | 106.7384* | 102.5174 | 103.8569 | 106.5072* |
| | | $\overline{y}_{KC2}$ | 104.9514 | 107.0019 | 111.8141* | 104.7593 | 106.4286 | 111.5613* |
| | | $\overline{y}_{KC3}$ | 101.9209 | 101.4717 | 102.8348* | 101.7908 | 101.0864 | 102.6503* |
| | | $\overline{y}_{KC4}$ | 103.6176 | 105.661 | 109.2936* | 103.4846 | 105.2929 | 109.1322* |
| | | $\overline{y}_{KC5}$ | 101.8907 | 101.4433 | 102.7955* | 101.8192 | 101.1344 | 102.7487* |

distribution was lower than in the symmetrical distribution. In both real data and simulation studies, it is an advantage in terms of the usability of the proposed estimator that the necessary conditions are provided for the estimator to be effective. Therefore, the most efficient estimator in all numerical studies is the proposed estimator. When the estimators were compared among themselves, it was seen that $\overline{y}_{UK5}$ was superior to the others. However, this estimator includes more population parameters of the auxiliary variable. If only the mean of the auxiliary variable is known, the $\overline{y}_{UK1}$ estimator can be used as a more effective alternative than other estimators in the literature.

For future study, examining the proposed estimator, under other sampling methods, such as systematic, stratified, or ranked set sampling, can be considered as in the SRS method. Alternatively, different ratio estimators based on Uk's redescending M-estimator can also be suggested.

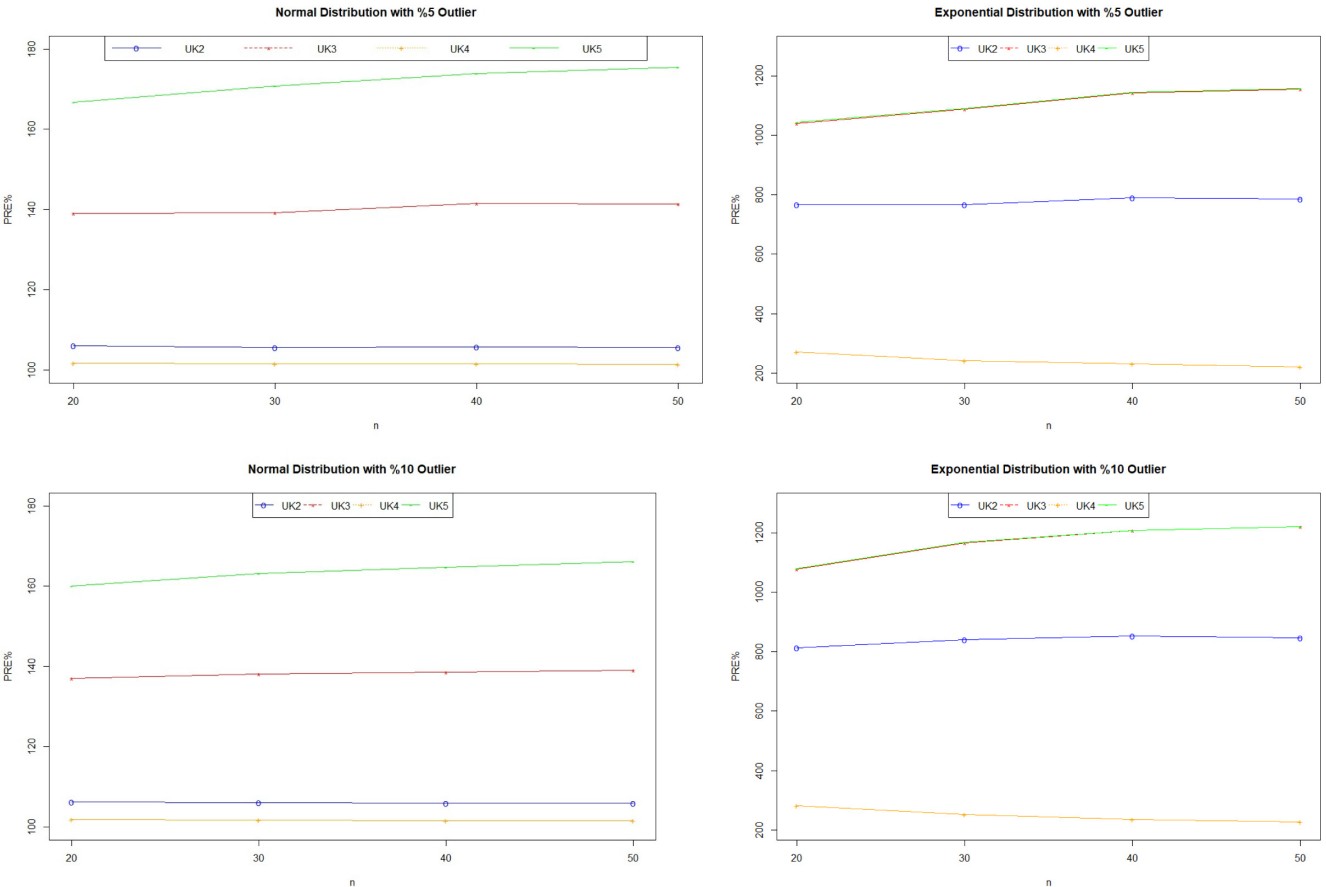

**Fig 5. PREs of the proposed estimator against $\overline{y}_{UK1}$ under symmetric and skewed distributions.**

## Acknowledgments

The Authors wish to thank the anonymous referee for the careful reading and constructive suggestions which led to improvement over an earlier version of the paper.

## Author Contributions

**Formal analysis:** Eda Gizem Koçyiğit, Ronald Onyango.

**Investigation:** Ronald Onyango, Cem Kadilar.

**Methodology:** Eda Gizem Koçyiğit.

**Project administration:** Ronald Onyango, Cem Kadilar.

**Resources:** Ronald Onyango.

**Validation:** Khalid Ul Islam Rather, Ronald Onyango.

**Visualization:** Ronald Onyango.

**Writing – review & editing:** Khalid Ul Islam Rather.

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
