## [Decision Letter · Decision Letter 0]

2 Nov 2022

PONE-D-22-27634Improved Regression in Ratio Type Estimators Based on Robust M-EstimationPLOS ONE

Dear Dr. RATHER,

Thank you for submitting your manuscript to PLOS ONE. After careful consideration, we feel that it has merit but does not fully meet PLOS ONE’s publication criteria as it currently stands. Therefore, we invite you to submit a revised version of the manuscript that addresses the points raised during the review process.

ACADEMIC EDITOR: The manuscript needs major revision. 

We look forward to receiving your revised manuscript.

Kind regards,

Nadia Hashim Al-Noor

Academic Editor

PLOS ONE

Journal Requirements:

2. Please note that PLOS ONE has specific guidelines on code sharing for submissions in which author-generated code underpins the findings in the manuscript. In these cases, all author-generated code must be made available without restrictions upon publication of the work. Please review our guidelines at https://journals.plos.org/plosone/s/materials-and-software-sharing#loc-sharing-code and ensure that your code is shared in a way that follows best practice and facilitates reproducibility and reuse. New software must comply with the Open Source Definition.

"NO" 

Reviewers' comments:

Reviewer's Responses to Questions

**Comments to the Author**

1. Is the manuscript technically sound, and do the data support the conclusions?

Reviewer #1: Partly

Reviewer #2: Yes

Reviewer #3: Partly

2. Has the statistical analysis been performed appropriately and rigorously? 

Reviewer #1: Yes

Reviewer #2: Yes

Reviewer #3: No

3. Have the authors made all data underlying the findings in their manuscript fully available?

Reviewer #1: Yes

Reviewer #2: Yes

Reviewer #3: Yes

4. Is the manuscript presented in an intelligible fashion and written in standard English?

Reviewer #1: Yes

Reviewer #2: Yes

Reviewer #3: Yes

5. Review Comments to the Author

Reviewer #1: To improve the novelty of the manuscript, some other objective functions should also be used.

The real data application to demonstrate the application of the proposal should be the part of the paper.

The overall language of the paper needs an improvement.

Reviewer #2: Referee report:

After reviewing the related literature in detail, authors develop a new family of the robust ratio estimators using the Uk’s redescending M-estimator successfully. In both application and simulation study, it is found that the proposed family is the most efficient estimator according to estimators in literature. As authors mention, these results support the theoretical efficiency condition obtained in (21). Authors derives two conditions from (21) but these conditions should be corrected as follows:

From (21),

B-B_UK+2R_j+B+B_UK-2b>0,

2B+2R_j-2b>0,

R_j>b-B.

Authors should evaluate the results of application and simulation based on the theoretical efficiency condition: R_j>b-B.

After revising the manuscript according to the mentioned correction, the manuscript can be published in Plos One.

Reviewer #3: The authors propose a new robust ratio type estimator using the Uk’s redescending M-estimator is proposed for the estimation of the finite population mean in the simple

24 random sampling (SRS) when there are outliers in the dataset. The properties of the proposed estimators are obtained and compared with the properties of the existing estimators. The theoretical results are enhanced empirically.

Although, theoretical and computational results are found satisfactory but, the paper is not suitable for publication in its current form. It needs revision under the comments highlighted in the pdf file attached in the attachment section.

6. PLOS authors have the option to publish the peer review history of their article (what does this mean?). If published, this will include your full peer review and any attached files.

Reviewer #1: No

Reviewer #2: No

Reviewer #3: No

---

## [Author Response · Author response to Decision Letter 0]

7 Nov 2022

Reviewer #1: To improve the novelty of the manuscript, some other objective functions should also be used. 

The real data application to demonstrate the application of the proposal should be the part of the paper. The overall language of the paper needs an improvement.

Dear Reviewer I ,

Thank you for your review and contributions. We started this work with the aim of developing the estimator proposed by Noor-ul Amin et al.(2020). The methods proposed before 2020 have already been compared in that article and it has been shown that the proposed ratio estimator is better than the estimators in the literature. We consider it sufficient to show that our proposed estimator is better than the proposed method in 2020. However, we expanded the Introduction section and gave the existing estimators. 

In our study, there are already real data studies in addition to the simulation study. If what you are talking about is solving numerical calculations step by step, this is quite difficult and complex due to objective functions. For this reason, we did all the operations on the R program and shared the codes with the journal.

We have worked on the changes you mentioned. We hope that the revised version of the work will satisfy you.

Regards.

Reviewer #2: After reviewing the related literature in detail, authors develop a new family of the robust ratio estimators using the Uk’s redescending M-estimator successfully. In both application and simulation study, it is found that the proposed family is the most efficient estimator according to estimators in literature. As authors mention, these results support the theoretical efficiency condition obtained in (21). Authors derives two conditions from (21) but these conditions should be corrected as follows:

From (21),

B-B_UK+2R_j+B+B_UK-2b>0,

2B+2R_j-2b>0,

R_j>b-B.

Authors should evaluate the results of application and simulation based on the theoretical efficiency condition: R_j>b-B.

After revising the manuscript according to the mentioned correction, the manuscript can be published in Plos One.

Dear Reviewer II ,

Thank you for your review and contributions. The expression "+" is misspelled in the formula in the equation you specified. The same equation has been written more clearly and corrected.

Regards.

Reviewer #3: The authors propose a new robust ratio type estimator using the Uk’s redescending M-estimator is proposed for the estimation of the finite population mean in the simple

24 random sampling (SRS) when there are outliers in the dataset. The properties of the proposed estimators are obtained and compared with the properties of the existing estimators. The theoretical results are enhanced empirically.

Although, theoretical and computational results are found satisfactory but, the paper is not suitable for publication in its current form. It needs revision under the comments highlighted in the pdf file attached in the attachment section.

Dear Reviewer III ,

Thanks for your careful review and advice. The corrections you requested have been made. We have made corrections based on your comments. The revised version of the paper has been submitted as a "Manuscript" file, and you can see changes with the highlighted version on the "Revised Manuscript with Track Changes" file your notes have been answered on “PONE-D-22-27634 response to comments” pdf file.

Regards.

---

## [Decision Letter · Decision Letter 1]

22 Nov 2022

PONE-D-22-27634R1

Improved Regression in Ratio Type Estimators Based on Robust M-Estimation

PLOS ONE

Dear Dr. RATHER,,

Thank you for submitting your manuscript to PLOS ONE. After careful consideration, we feel that it has merit but does not fully meet PLOS ONE’s publication criteria as it currently stands. Therefore, we invite you to submit a revised version of the manuscript that addresses the points raised during the review process.

ACADEMIC EDITOR: Revise your manuscript according to reviewers comments. Also, revise your introduction section by mentioning the latest related publish papers. The following papers can also be discussed.ShahzadU,AlnoorNH,HanifM,SajjadI,AnasMM.Imputationbasedmeanestimatorsincaseofmissing data utilizing robust regression and variance-covariance matrices. Communications in StatisticsSimulation and Computation. 2020a. 32. ShahzadU,AlnoorNH,HanifM,SajjadI,AnasMM.Quantileregression-ratio-typeestimators for mean estimation under complete and partial auxiliary information. Scientia Iranica. 2020b. https://doi.org/10.24200/sci.2020.54423.3744/

We look forward to receiving your revised manuscript.

Kind regards,

Academic Editor

PLOS ONE

Journal Requirements:

<p

Reviewers' comments:

Reviewer's Responses to Questions

**Comments to the Author**

1. If the authors have adequately addressed your comments raised in a previous round of review and you feel that this manuscript is now acceptable for publication, you may indicate that here to bypass the “Comments to the Author” section, enter your conflict of interest statement in the “Confidential to Editor” section, and submit your "Accept" recommendation.

Reviewer #2: All comments have been addressed

Reviewer #3: (No Response)

2. Is the manuscript technically sound, and do the data support the conclusions?

Reviewer #2: Yes

Reviewer #3: Partly

3. Has the statistical analysis been performed appropriately and rigorously? 

Reviewer #2: Yes

Reviewer #3: Yes

4. Have the authors made all data underlying the findings in their manuscript fully available?

Reviewer #2: Yes

Reviewer #3: Yes

5. Is the manuscript presented in an intelligible fashion and written in standard English?

Reviewer #2: Yes

Reviewer #3: Yes

6. Review Comments to the Author

Reviewer #2: Manuscript is well now. All the changes are correct as per suggestion, can be published in your reputed journal PLOS.

Reviewer #3: Review report

Title: Improved regression in ratio type estimators based on robust M-estimation

The authors have successfully incorporated the comments; however, the introduction section needs to be revised by incorporating the latest related references published till date. The following references can also be added to enhance the readability of the paper.

Bhushan, S. and Kumar, A. (2022). Novel log type class of estimators under ranked set sampling. Sankhya B, 84, 421-447. https://doi.org/10.1007/s13571-021-00265-y

Bhushan, S., Kumar, A., Shahab, S., Lone, S.A. and Almutlak, S.A. (2022). Modified class of estimators using ranked set sampling. Mathematics, 10, 3921, 1-13

Bhushan, S., Kumar, A. and Lone, S.A. (2022). On some novel classes of estimators under ranked set sampling. AEJ-Alexandria Engineering Journal, 61, 5465-5474. https://doi.org/10.1016/j.aej.2021.11.001.

Bhushan, S., Kumar, A., Pandey, A.P. and Singh, S. (2022). Estimation of population mean in presence of missing data under simple random sampling. Communications in Statistics - Simulation and computation. https://doi.org/10.1080/03610918.2021.2006713

Bhushan, S., Kumar, A. and Singh, S. (2021). Some efficient classes of estimators under stratified sampling. Communications in Statistics - Theory and Methods, 1-30. DOI:10.1080/03610926.2021.1939052.

Bhushan, S., Kumar, A., Akhtar, M.T. and Lone. S.A. (2022). Logarithmic type predictive estimators under simple random sampling. AIMS Mathematics, 7(7), 11992-12010.

Bhushan, S., Kumar, A., Shahab, S., Lone. S.A. and Akhtar, M.T. (2022). On efficient estimation of population mean under stratified ranked set sampling. Journal of Mathematics, 2022(3), 1-20.

Bhushan, S., Kumar, A., Onyango, R. and Singh, S. (2022). Some improved classes of estimators in stratified sampling using bivariate auxiliary information. Journal of Probability and Statistics, 2022(2), 1-23.

Bhushan, S., Kumar, A., Singh, S. and Kumar, S. (2021). An improved class of estimators of population mean under simple random sampling. Philippine Statistician, 70(1), 33-47.

7. PLOS authors have the option to publish the peer review history of their article (what does this mean?). If published, this will include your full peer review and any attached files.

Reviewer #2: No

Reviewer #3: No

While revising your submission, please upload your figure files to the Preflight Analysis and Conversion Engine (PACE) digital diagnostic tool, https://pacev2.apexcovantage.com/. PACE helps ensure that figures meet PLOS requirements. To use PACE, you must first register as a user. Registration is free. Then, login and navigate to the UPLOAD tab, where you will find detailed instructions on how to use the tool. If you encounter any issues or have any questions when using PACE, please email PLOS at figures@plos.org. Please note that Supporting Information files do not need this step.</p

---

## [Author Response · Author response to Decision Letter 1]

23 Nov 2022

RESPONSE TO REVIEWERS

Reviewer #2: Manuscript is well now. All the changes are correct as per suggestion, can be published in your reputed journal PLOS.

Dear Reviewer II,

Thank you for your careful review and valuable contribution to our paper in this process.

Regards.

Reviewer #3: The authors have successfully incorporated the comments; however, the introduction section needs to be revised by incorporating the latest related references published till date. The following references can also be added to enhance the readability of the paper.

Bhushan, S. and Kumar, A. (2022). Novel log type class of estimators under ranked set sampling. Sankhya B, 84, 421-447. https://doi.org/10.1007/s13571-021-00265-y

Bhushan, S., Kumar, A., Shahab, S., Lone, S.A. and Almutlak, S.A. (2022). Modified class of estimators using ranked set sampling. Mathematics, 10, 3921, 1-13

Bhushan, S., Kumar, A. and Lone, S.A. (2022). On some novel classes of estimators under ranked set sampling. AEJ-Alexandria Engineering Journal, 61, 5465-5474. https://doi.org/10.1016/j.aej.2021.11.001.

Bhushan, S., Kumar, A., Pandey, A.P. and Singh, S. (2022). Estimation of population mean in presence of missing data under simple random sampling. Communications in Statistics - Simulation and computation. https://doi.org/10.1080/03610918.2021.2006713

Bhushan, S., Kumar, A. and Singh, S. (2021). Some efficient classes of estimators under stratified sampling. Communications in Statistics - Theory and Methods, 1-30. DOI:10.1080/03610926.2021.1939052.

Bhushan, S., Kumar, A., Akhtar, M.T. and Lone. S.A. (2022). Logarithmic type predictive estimators under simple random sampling. AIMS Mathematics, 7(7), 11992-12010.

Bhushan, S., Kumar, A., Shahab, S., Lone. S.A. and Akhtar, M.T. (2022). On efficient estimation of population mean under stratified ranked set sampling. Journal of Mathematics, 2022(3), 1-20.

Bhushan, S., Kumar, A., Onyango, R. and Singh, S. (2022). Some improved classes of estimators in stratified sampling using bivariate auxiliary information. Journal of Probability and Statistics, 2022(2), 1-23.

Bhushan, S., Kumar, A., Singh, S. and Kumar, S. (2021). An improved class of estimators of population mean under simple random sampling. Philippine Statistician, 70(1), 33-47.

Dear Reviewer III,

Thank you for your careful review and valuable contribution to our paper in this process. The latest related references you mentioned in your comments have been added to the "Introduction" section of the manuscript.

Regards.

RESPONSE TO ACADEMIC EDITOR

ACADEMIC EDITOR: Revise your manuscript according to reviewers comments. Also, revise your introduction section by mentioning the latest related publish papers. The following papers can also be discussed.

Shahzad U, Alnoor NH, Hanif M, Sajjad I, Anas MM. Imputation based mean estimators in case of missing data utilizing robust regression and variance-covariance matrices. Communications in Statistics Simulation and Computation. 2020a. 32. 

Shahzad U, Alnoor NH, Hanif M,Sajjad I,Anas MM. Quantileregression-ratio-type estimators for mean estimation under complete and partial auxiliary information. Scientia Iranica. 2020b. https://doi.org/10.24200/sci.2020.54423.3744/

Thank you for your contribution to our paper. We revised the manuscript according to the reviewers' comments. Also, the cited references you pointed out have been added to the "Introduction" section.

Regards.

---

## [Editor Report · Decision Letter 2]

28 Nov 2022

Improved Regression in Ratio Type Estimators Based on Robust M-Estimation

PONE-D-22-27634R2

Dear Mr. Rather,

We’re pleased to inform you that your manuscript has been judged scientifically suitable for publication and will be formally accepted for publication once it meets all outstanding technical requirements.

Kind regards,

Nadia Hashim Al-Noor, Ph.D.

Academic Editor

PLOS ONE
---

## [Editor Report · Acceptance letter]

1 Dec 2022

PONE-D-22-27634R2 

Improved regression in ratio type estimators based on robust M-estimation 

Dear Dr. Rather:

I'm pleased to inform you that your manuscript has been deemed suitable for publication in PLOS ONE. Congratulations! Your manuscript is now with our production department. 

Kind regards, 

on behalf of

Dr. Nadia Hashim Al-Noor 

Academic Editor

PLOS ONE